# Association of Social Capital and Locus of Control with Perceived Health during the COVID-19 Pandemic in Japan

**DOI:** 10.3390/ijerph19159415

**Published:** 2022-08-01

**Authors:** Mitsuru Mori, Toshiaki Seko, Shunichi Ogawa

**Affiliations:** Department of Health Science, Hokkaido Chitose College of Rehabilitation, 2-10, Satomi, Chitose 066-0055, Japan; m-mori@chitose-reha.ac.jp (M.M.); s-ogawa@chitose-reha.ac.jp (S.O.)

**Keywords:** social capital, locus of control, COVID-19, perceived health

## Abstract

Previous studies have indicated that social capital and locus of control influence mental health. Accordingly, we investigated the effect of social capital and locus of control on perceived physical and mental health in the general Japanese population during the COVID-19 pandemic. In order to conduct a cross-sectional study, in 2021, three thousand citizens were randomly selected from the Chitose City Resident Register according to ten strata of sex and age classes between 30 years and 79 years. Because thirteen persons moved away from the city, the survey was conducted for the remaining 2987 citizens. A total of 1430 citizens (712 males, 718 females) responded to the survey with their written informed consent (response rate, 47.9%). As a result, social capital measured three dimensions, namely social support, social participation, and trust and reciprocity, and internal locus of control was significantly inversely associated with it, but external locus of control was significantly positively associated with impaired physical and mental health in male and female subjects after adjustment of lifestyle habits and lifestyle change affected by the pandemic. Strengthening social capital and internal locus of control, and weakening external locus of control, may improve physical and mental health, even if the pandemic would bring about distress. Further longitudinal study is needed to examine the causal relationship among them.

## 1. Introduction

After the first occurrence of COVID-19 in China in December 2019, the spread of the disease caused a worldwide pandemic, including in Japan. At the end of June 2022, there have been more than nine million cases and more than thirty thousand deaths attributed to COVID-19 in Japan. Most surveys of the general population show increased symptoms of depression, anxiety, and stress related to COVID-19, as a result of psychological stressors such as life disruption, fear of illness, or fear of negative economic effects [1]. During the pandemic in Japan, reduced physical activity has been reported [2,3,4], and other deteriorations in lifestyle, such as insufficient sleep and nutritious meals related to psychological distress, have been also shown [5]. Increased psychological distress during the COVID-19 pandemic has been reported in individuals, especially those with lower social economic status [5,6], as well as healthcare workers in Japan [7].

Despite the same stress and fear affecting the entire population, an individual’s perception of the degree of stress can vary. Several important factors influence a person’s perceived level of stress, with social capital and locus of control being studied before and during the pandemic. Social capital has been broadly defined as resources emerging from networks of trust, but there is currently no generally accepted instrument for measuring social capital. Nieminen et al. [8] measured social capital using variables that constitute three dimensions, such as social support, social participation, and trust and reciprocity, and, according to their cross-sectional study in Finland between 2010 and 2011, a high level of social capital is associated with good self-rated health.

Several studies have indicated that social capital has influenced mental health during the COVID-19 pandemic [9,10,11,12,13,14]. For instance, Budimir et al. [14] indicated, from a cross-sectional study in Austria during the pandemic, that a higher level of social support was negatively associated with perceived stress, depression, anxiety, and insomnia.

Locus of control was firstly developed by Rotter [15], consisting of two different types of beliefs, such as internal and external locus of control. Internal locus of control was described as the degree of an individual’s belief that they have personal control over the outcome of a life event. By contrast, external locus of control was described as the degree of an individual’s belief that external forces such as luck, destiny, and powerful others have control over the outcomes of life events. A cross-sectional study in Japan in 2006 showed that a higher internal locus of control was inversely associated with an increased risk of depression, but a higher external locus of control was positively associated with an increased risk of depressive symptoms [16].

Several studies have indicated that locus of control has influenced mental health during the COVID-19 pandemic [17,18,19,20]. For instance, Krampe et al. [20] showed, from a cross-sectional study in Norway and Germany during the pandemic, that a higher level of internal locus of control was negatively associated with general mental distress, but an external locus of control was positively associated with general mental distress.

The aim of this study was to investigate the effect of social capital and locus of control on perceived physical and mental health in the general Japanese population during the pandemic.

## 2. Materials and Methods

### 2.1. Procedure

In order to conduct a cross-sectional study, three thousand citizens were randomly selected from the Chitose City Resident Register according to ten strata of sex and age classes between 30 years and 79 years in September 2021. Each stratum from the age of thirties to seventies by sex included three hundred randomly selected citizens. Thereafter, written informed consent was obtained from each participant, and the survey was conducted from October to November 2021. The study was approved by the Committee of the Institutional Review Board of Hokkaido Chitose College of Rehabilitation (R02101). This study was supported in part by JSPS KAKENHI, Grant Number JP21K10476. The authors declare no conflicts of interest.

### 2.2. Survey Components

Almost all question variables were binary, and zero or one was assigned for each answer, and they were summed in each item of the objective and explanatory variables.

#### 2.2.1. Impaired Physical and Mental Health

Both perceived physical and mental health were selected as the objective variables. Some questions were chosen from the the Center for Epidemiological Survey—Depression (CES-D) scale developed by Radloff [21]. Impaired physical health was determined by the following six options: During the past week, (i) My health status was good (no = 1); (ii) I did not feel like eating, and my appetite was poor (yes = 1); (iii) My sleep was restless (yes = 1); (iv) I felt pain in my body (yes = 1); (v) I felt dizzy and/or had tinnitus (yes = 1); (vi) I tended to be tired (yes = 1).

Impaired mental health was determined by the following seven options: During the past week, (i) I enjoyed life (no = 1); (ii) I felt depressed (yes = 1); (iii) I was happy (no = 1); (iv) I felt sad (yes = 1); (v) I felt hopeful about the future (no = 1); (vi) I was bothered by things that usually don’t bother me (yes = 1); (vii) I spoke less than usual (yes = 1).

#### 2.2.2. Social Capital

Social capital was selected as a part of the explanatory variable, and was measured with three dimensions, such as social support, social participation, and trust and reciprocity, with reference to the study conducted by Nieminen et al. [8]. Social support was determined by the following two statements: (i) I have someone who will help me when I am troubled (yes = 1); (ii) I have someone who makes me feel better when I feel down (yes = 1). Social participation was covered by the following four statements: (i) I engage in physical activities with other people (yes = 1); (ii) I engage in cultural activities with other people (yes = 1); (iii) I engage in voluntary societies (yes = 1); (iv) I have an occupation (yes = 1). Trust and reciprocity were determined in the form of the following two statements: (i) I usually think that I will help my friend or neighbor when he/she is troubled (yes = 1); (ii) I usually think that I will seek help from my friend or neighbor when I am troubled (yes = 1).

#### 2.2.3. Locus of Control

Locus of control was selected as another part of the explanatory variable, and internal and external locus control were measured with reference to the questionnaire developed by Rotter [15]. Internal locus of control was determined with the following five statements: I usually think that (i) My sincere talk allowed everybody to understand me (yes = 1); (ii) I determine my fate by myself (yes = 1); (iii) Becoming happy or unhappy is a matter of hard work (yes = 1); (iv) My strong effort leads me to greater achievement (yes = 1); (v) My own decision leads me to better results (yes = 1). External locus of control was determined with the following three options: I usually think that (i) It seems better to let things take their own course (yes = 1); (ii) Chance or luck plays an important role in my life (yes = 1); (iii) Becoming happy or unhappy is a matter of chance (yes = 1).

#### 2.2.4. Potential Confounders

Lifestyle habits were adjusted as potential confounders in the analysis. Preferable lifestyle habits were measured with the following eight statements: (i) I perform physical activity more than 20 min. per day, or two hours per week (yes = 1); (ii) I sleep more than seven hours every night (yes = 1); (iii) I eat breakfast almost every morning (yes = 1); (iv) I keep in mind that I should eat vegetables and fruits (yes = 1); (v) I keep in mind that I should eat fish (yes = 1); (vi) I maintain my BMI (body mass index), which was calculated from body weight (kg) divided by square of body height (m), between 18.5 and 25.0 (yes = 1); (vii) I smoke tobacco (no = 1), (viii) I drink alcoholic beverages more than 3 times per week (no = 1).

Lifestyle changes affected by the COVID-19 pandemic were also adjusted as potential confounders in the analysis, and were measured with the following five statements: During COVID-19 pandemic, (i) I prefer to stay home, doing self-reflection to refrain from going outside (yes = 1); (ii) I prefer to stay home, because there is no school for my child or grandchild (yes = 1); (iii) The pandemic increased the burden in the workplace (yes = 1); (iv) The pandemic forced me to do telework at home (yes = 1); (v) The pandemic regrettably reduced my workload (yes = 1).

Eight aspects of past medical history, such as hypertension, diabetes mellitus, heart disease, apoplexy, cancer, kidney disease, and lung disease, were also surveyed and adjusted as potential confounders in the analysis.

### 2.3. Statistical Methods

We divided the study subjects into four groups according to sex and age classes. The median age (60.5 years for the male subjects, and 54.3 years for the female subjects) was used to divide them into two age classes by sex. The Kruskal–Wallis test was used for examining the difference in the objective and explanatory variables among the four groups. Multiple linear regression analysis was conducted for examining the association between the objective and explanatory variables among the four groups, adjusting potential confounding variables such as age, lifestyle habits, lifestyle changes affected by COVID-19 pandemic, and past medical history. SAS statistical software was used for all analyses (SAS version 9.4, SAS Institute, Tokyo, Japan). The NPAR1WAY and REG procedures in SAS were utilized for the Kruskal–Wallis test and the multiple linear regression analysis, respectively [22,23]. Significance level was set at probability of 0.05.

## 3. Results

Among the three thousand randomly selected citizens, thirteen persons moved away from the city before the survey. Therefore, the survey was conducted for the remaining 2987 citizens, and 1430 citizens (712 males, 718 females) responded to the survey with their written informed consent (response rate, 47.9%). The average age of the respondents was 56.2 years (standard deviation, 14.1), and the age range was between 30 and 79 years.

Table 1 presents the distribution of objective and explanatory variables according to the four groups by sex and age classes. As a result of the Kruskal–Wallis test, differences among the four groups were significantly observed in the variables of impaired physical health, with the highest mean in the older female subjects (*p* = 0.001), impaired mental health, with the highest mean in the younger female subjects (*p* < 0.001), social support, with the highest mean in the younger female subjects (*p* < 0.001), social participation, with the highest mean in the younger male subjects (*p* = 0.002), and trust and reciprocity, with the highest mean in the younger female subjects (*p* < 0.001), but a significant difference was not observed in internal or external locus of control.

Table 2 shows the results of the multiple linear regression analysis performed with impaired physical health as the objective variable and social capital and locus of control as the explanatory variables, according to the four groups of sex and age classes. Age, lifestyle habits, lifestyle changes affected by COVID-19 pandemic, and past medical history were simultaneously included in the model for adjustment.

Social support was significantly inversely associated with impaired physical health in the younger male subjects (adjusted regression coefficient, or β = −0.351, *p* = 0.003) and the older male subjects (β = −0.254, *p* = 0.016). Social participation was significantly inversely associated with impaired physical health in the younger male subjects (β = −0.322, *p* = 0.002), the older male subjects (β = −0.143, *p* = 0.021), and the younger female subjects (β = −0.244, *p* = 0.012).

Internal locus of control was significantly inversely associated with impaired physical health in the older male subjects (β = −0.152, *p* = 0.009) and the younger female subjects (β = −0.199, *p* = 0.002). External locus of control was significantly positively associated with impaired physical health in the younger male subjects (β = 0.210, *p* = 0.010) and the older female subjects (β = 0.177, *p* = 0.039).

Table 3 shows the results of the multiple linear regression analysis performed with impaired mental health as the objective variable and social capital and locus of control as the explanatory variables, according to the four groups of sex and age classes. Age, lifestyle habits, lifestyle changes affected by COVID-19 pandemic, and past medical history were simultaneously included in the model for adjusting them as well.

Social support was significantly inversely associated with impaired mental health in the younger male subjects (β = −0.837, *p* < 0.001), the older male subjects (β = −0.658, *p* < 0.001), the younger female subjects (β = −0.946, *p* < 0.001), and the older female subjects (β = −0.844, *p* < 0.001). Social participation was significantly inversely associated with impaired mental health in the younger male subjects (β = −0.631, *p* < 0.001), the older male subjects (β = −0.363, *p* < 0.001), and the older female subjects (β = −0.244, *p* = 0.028). Trust and reciprocity was significantly inversely associated with impaired mental health in the younger male subjects (β = −0.670, *p* < 0.001).

Internal locus of control was significantly inversely associated with impaired mental health in the younger male subjects (β = −0.424, *p* < 0.001), the older male subjects (β = −0.414, *p* < 0.001), the younger female subjects (β = −0.476, *p* < 0.001), and the older female subjects (β = −0.468, *p* < 0.001). External locus of control was significantly positively associated with impaired mental health in the younger male subjects (β = 0.454, *p* < 0.001), the older male subjects (β = 0.363, *p* < 0.001), and the older female subjects (β = 0.244, *p* = 0.038).

## 4. Discussion

From the results of the cross-sectional study during the COVID-19 pandemic in Japan, we found that social capital measured with three dimensions, such as social support, social participation, and trust and reciprocity, and internal locus of control were inversely associated, while external locus of control was positively associated, with impaired physical and mental health in male and female subjects, even after the adjustment of lifestyle habits and lifestyle changes affected by the pandemic. Females showed relatively higher impairments in physical and mental health than males, and such female dominance in health impairment is consistent with other studies [9,24]. Although the findings were not significant in all of the four groups of sex and age classes, the direction of positive or negative association was consistent among them.

Significant inverse associations between higher social capital and impaired mental health have been reported during the pandemic [9,10,11,12,13,14]. However, the significant inverse association between higher social capital and impaired physical health in addition to impaired mental health was unique to our results. Grey et al. [9] revealed, from a cross-sectional study in Lebanon during the pandemic, that high social support was a protective factor for depression and poor sleep quality. Sun et al. [10] showed, from a cross-sectional study in Shanghai, China during the pandemic, that social capital played an important role in improving deteriorated mental health, including depressive status. Caballero-Dominguez et al. [11] indicated, from a cross-sectional study in Colombia during the pandemic, that low cognitive social capital was associated with increased risk of depression, suicide, perceived stress, and insomnia. Cognitive social capital is compatible with trust and reciprocity in our study in concept.

Li et al. [12] reported, from a cross-sectional study in Hong Kong, China during the pandemic, that a lack of cognitive social capital, such as interpersonal trust, social harmony, and belonging, was associated with increased risk of depression. Sato et al. [13] reported, from a longitudinal study in Japan during the pandemic, that social cohesion and reciprocity was negatively associated with an increased risk of depressive symptoms.

Significant associations between locus of control and impaired mental health have been also reported during the pandemic [17,18,19,20]. However, the significant association between locus of control and impaired physical health in addition to impaired mental health was unique to our results, too. Flesia et al. [17] showed, from a cross-sectional study in Italy during the pandemic, that a higher level of internal locus of control was associated with a lower level of perceived stress. Sigurvinsdottir et al. [18] indicated, from a cross-sectional study in the United States and five European countries during the pandemic, that a higher level of internal locus of control was inversely associated with depression symptoms and anxiety symptoms, but a higher level of external locus of control, namely chance locus of control and powerful others’ locus of control, was positively associated with depression symptoms and anxiety symptoms. Alat et al. [19] reported, from a cross-sectional study in India during the pandemic, that a higher level of internal locus of control was negatively associated with psychological distress.

There are several limitations in this study. Firstly, our survey, with a random sampling from citizens in the community, was desirable for reducing selection bias. However, because the response rate was not so high (47.9%), selection bias may exist in our study. Secondly, data on some potential confounding factors, such as socio-economic status, were not included in this study, although age, lifestyle habits, lifestyle changes affected by the COVID-19 pandemic, and past medical history were adjusted in the analysis. Socio-environmental status such as marital status, number of family members in the household, any family members having COVID-19 infection, access to health facilities, and the type of occupation should be included in the questionnaire in a future survey. Thirdly, although we focused on and surveyed social capital and locus of control as personal characteristics in our study, other components of personal characteristics, such as resilience and coping, are not included. Fourthly, because the study design was a cross-sectional study, a causal relationship could not be indicated. A longitudinal study with a larger sample size is needed to examine the causal relationship among them in the future.

## 5. Conclusions

Our findings can offer important insights to overcome the distress experienced during the pandemic. Policymaking for the COVID-19 pandemic may focus on strengthening social capital and internal locus of control, and weakening external locus of control may improve physical and mental health. Workers, especially essential and/or healthcare workers, whose perceived health might have been negatively affected by the pandemic, are most recommended for informing evidence about the association of social capital and locus of control with physical and mental health.

## Figures and Tables

**Table 1 ijerph-19-09415-t001:** Distribution of objective and explanatory variables according to the four groups by sex and age classes ^$^.

Item	Group by Sex and Age Class	Number	Mean	Standard Deviation	Median	Kruskal–Wallis Test
Objective variables						
Impaired physical health	Younger male subjects	353	1.67	1.42	2	*p* = 0.001
	Older male subjects	350	1.67	1.66	1	
	Younger female subjects	356	1.92	1.42	2	
	Older female subjects	348	2.03	1.48	2	
	Total	1407	1.82	1.43	2	
Impaired mental health	Younger male subjects	350	2.05	2.10	1	*p* < 0.001
	Older male subjects	345	1.28	1.66	1	
	Younger female subjects	355	2.06	2.07	1	
	Older female subjects	342	2.03	1.48	2	
	Total	1392	1.81	2.00	1	
Explanatory variables						
Social support	Younger male subjects	356	1.68	0.64	2	*p* < 0.001
	Older male subjects	352	1.65	0.68	2	
	Younger female subjects	357	1.87	0.43	2	
	Older female subjects	356	1.81	0.53	2	
	Total	1421	1.75	0.58	2	
Social participation	Younger male subjects	357	1.40	0.73	1	*p* = 0.002
	Older male subjects	352	1.25	0.94	1	
	Younger female subjects	357	1.23	0.80	1	
	Older female subjects	356	1.22	1.04	1	
	Total	1422	1.27	0.89	1	
Trust and reciprocity	Younger male subjects	357	1.30	0.60	1	*p* < 0.001
	Older male subjects	352	1.34	0.57	1	
	Younger female subjects	357	1.57	0.55	2	
	Older female subjects	359	1.49	0.58	2	
	Total	1425	1.43	0.59	1	
Internal locus of control	Younger male subjects	351	2.48	1.24	2	*p* = 0.313
	Older male subjects	344	2.57	1.26	3	
	Younger female subjects	352	2.58	1.21	3	
	Older female subjects	344	2.46	1.26	2	
	Total	1391	2.52	1.24	3	
External locus of control	Younger male subjects	354	0.78	0.93	1	*p* = 0.053
	Older male subjects	351	0.83	0.99	0	
	Younger female subjects	354	0.77	0.86	1	
	Older female subjects	349	0.93	0.92	1	
	Total	1408	0.83	0.93	1	

^$^: The age classes were created using the median age by sex (60.5 years and 54.3 years for the male and female subjects, respectively).

**Table 2 ijerph-19-09415-t002:** Adjusted ^#^ regression coefficient (β) with its standard error (SE) on impaired physical health regarding social capital and locus of control for the four groups of objective and explanatory variables according to the four groups by sex and age classes ^$^.

Item	Group by Sex and Age Class	β	SE	*t*-Value	Significance Level
Social support	Younger male subjects	−0.351	0.115	−3.04	0.003
	Older male subjects	−0.254	0.150	−2.42	0.016
	Younger female subjects	−0.308	0.179	−1.72	0.086
	Older female subjects	−0.250	0.155	−1.62	0.106
Social participation	Younger male subjects	−0.322	0.101	−3.19	0.002
	Older male subjects	−0.143	0.062	−2.32	0.021
	Younger female subjects	−0.244	0.096	−2.53	0.012
	Older female subjects	−0.058	0.082	−0.72	0.475
Trust and reciprocity	Younger male subjects	−0.169	0.124	−1.36	0.174
	Older male subjects	0.060	0.129	0.05	0.644
	Younger female subjects	−0.041	0.138	−0.30	0.768
	Older female subjects	−0.196	0.139	−1.40	0.161
Internal locus of control	Younger male subjects	−0.086	0.060	−1.43	0.152
	Older male subjects	−0.152	0.058	−2.64	0.009
	Younger female subjects	−0.199	0.063	−3.17	0.002
	Older female subjects	−0.121	0.065	−1.87	0.063
External locus of control	Younger male subjects	0.210	0.081	2.59	0.010
	Older male subjects	0.049	0.073	0.68	0.498
	Younger female subjects	0.169	0.087	1.94	0.053
	Older female subjects	0.177	0.085	2.07	0.039

^#^: Adjusted for age, lifestyle habits, lifestyle changes affected by the COVID-19 pandemic, and past medical history. ^$^: The age classes were created using the median age by sex (60.5 years and 54.3 years for the male and female subjects, respectively).

**Table 3 ijerph-19-09415-t003:** Adjusted ^#^ regression coefficient (β) with its standard error (SE) on impaired mental health regarding social capital and locus of control for the four group of objective and explanatory variables according to the four groups by sex and age classes ^$^.

Item	Group by Sex And Age Class	β	SE	*t*-Value	Significance Level
Social support	Younger male subjects	−0.837	0.165	−5.08	<0.001
	Older male subjects	−0.658	0.124	−5.29	<0.001
	Younger female subjects	−0.948	0.255	−3.71	<0.001
	Older female subjects	−0.844	0.207	−4.07	<0.001
Social participation	Younger male subjects	−0.631	0.147	−4.29	<0.001
	Older male subjects	−0.363	0.093	−3.89	<0.001
	Younger female subjects	−0.264	0.140	−1.91	0.057
	Older female subjects	−0.244	0.110	−2.21	0.028
Trust and reciprocity	Younger male subjects	−0.670	0.180	−3.73	<0.001
	Older male subjects	0.024	0.158	0.15	0.881
	Younger female subjects	−0.331	0.199	−1.66	0.097
	Older female subjects	−0.339	0.188	−1.80	0.073
Internal locus of control	Younger male subjects	−0.424	0.086	−4.92	<0.001
	Older male subjects	−0.414	0.069	−6.03	<0.001
	Younger female subjects	−0.476	0.088	−5.40	<0.001
	Older female subjects	−0.468	0.087	−5.38	<0.001
External locus of control	Younger male subjects	0.454	0.117	3.88	<0.001
	Older male subjects	0.363	0.088	4.15	<0.001
	Younger female subjects	0.229	0.126	1.82	0.070
	Older female subjects	0.244	0.117	2.09	0.038

^#:^ Adjusted for age, lifestyle habits, lifestyle changes affected by the COVID-19 pandemic, and past medical history. ^$^: The age classes were created using the median age by sex (60.5 years and 54.3 years for the male and female subjects, respectively).

## Data Availability

Not applicable.

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
