# Peer review of "Association of Social Capital and Locus of Control with Perceived Health during the COVID-19 Pandemic in Japan"

_ijerph, 2022, doi:10.3390/ijerph19159415_

Round 1

Reviewer 1 Report

The study was designed to assess the associations among locus control, social capital and perceived mental and physical health during Covid-19 pandemic. However, similar studies have been conducted and reported before in many countries. 

Major comment:  Several essential factors that contribute to the perceived health are not considered in the analysis and authors must have them. For example, marital status of the participants, size of the family, how many family members were living in the household, any family members had covid-19,  how many covid-19 infections a participant had during the pandemic, and if the participant has easy access to a health facilities? In addition, the occupation of the participant should be considered, people of certain occupations are more resilient and able to handle the stress better. 

Author Response

Response to the comments by Reviewer 1.

As Reviewer 1 pointed out, we agree with their comment that the factors associated with socioeconomic status influence the perceived health. However, we did not include these factors in the questionnaire, because participants might be sensitive to them, and we were concerned that such questions could reduce a response rate. Although we have already stated briefly in the section of the second limitation in Discussion, we add the following sentences in Discussion.

“Socio-environmental status such as marital status, size of the family members in the household, any family members having COVID-19 infection, access to health facilities, and the kind of occupation should be included in the questionnaire in the future survey.”

Reviewer 2 Report

Abstract: please re-work abstract, the reason why you study social capital and locus just appear in the end of abstract and it is not sufficiently justify.

INTRODUCTION: l.25-33: please bring some datas, statistical reality, to provid solid evidence.

Introduction: please explain why locus and social capital elements related to stress. Your work is not enough justify.

Introduction need more reference to present factors of stress

Introduction: please add example to illustrate locus and explain why it leads to stress/depression (in Japan).

The paper need to be improve about why you choose locus and social capital.

Author Response

Response to the comments by Reviewer 2.

  1. Abstract: The following sentence was added in Abstract, according to the comment by Reviewer 2.

“Previous studies have indicated that social capital and locus of control influenced mental health. Accordingly, we investigated the effect of social capital and locus of control on perceived physical and mental health in the general Japanese population during the COVID-19 pandemic.”

  1. Introduction: According to the comments by the Reviewers 2, 3, & 4, a total of ten references were additionally cited in Introduction, which were cited in Discussion of the previous manuscript. The following sentences were added in Introduction.

“Several studies have indicated that social capital influenced mental health during the COVID-19 pandemic.9-14 For instance, Budimir et al.14 indicated from a cross-sectional study in Austria during the pandemic that a higher level of social support was negatively associated with perceived stress, depression, anxiety, and insomnia.”

“Several studies have indicated that locus of control influenced mental health during the COVID-19 pandemic.17-20 For instance, Krampe et al.20 showed from a cross-sectional study in Norway and Germany during the pandemic that a higher level of internal locus of control was negatively associated with general mental distress, but an external locus of control was positively associated with general mental distress.”

Reviewer 3 Report

The study is highly interesting, not only because it shades light on current disruptions in personal and organizational routines, but it could be relevant at macro-level, as well as organizational level for better adaptation of employee strategies. We recommend to include the study in a future issue, having in mind a few aspects to be incorporated in the final version of the paper. 

The introduction is well balanced. It briefly presents the main concepts used, as well as relevant studies for the topic investigated. Nevertheless, we recommend to introduce a section dedicated to more in-depth analysis of the academic framework - both considering the literature review pinpointing the points of reference for social capital and locus of control (necessary for a more elaborated discussion of the findings), and the previous studies (also useful for complementing the results of the present study). 

Discussions (or the conclusions) section might take into considering to provide some recommendations / implications relevant at macro-level to help policy-makers to better deal with the aftermaths of the pandemic. Also, at organizational levels implications should be discussed to guide managers cope with the changes in the workforce. 

Author Response

Response to the comments Reviewer 3.

  1. Introduction: According to the comments by the Reviewers 2, 3, & 4, a total of ten references were additionally cited in Introduction, which were cited in Discussion of the previous manuscript. The following sentences were added in Introduction,

“Several studies have indicated that social capital influenced mental health during the COVID-19 pandemic.9-14 For instance, Budimir et al.14 indicated from a cross-sectional study in Austria during the pandemic that a higher level of social support was negatively associated with perceived stress, depression, anxiety, and insomnia.”

“Several studies have indicated that locus of control influenced mental health during the COVID-19 pandemic.17-20 For instance, Krampe et al.20 showed from a cross-sectional study in Norway and Germany during the pandemic that a higher level of internal locus of control was negatively associated with general mental distress, but an external locus of control was positively associated with general mental distress.”

  1. Conclusions: Although we have already stated briefly some recommendations or implications in Conclusions, the following sentence was added in Conclusions, according to the comments by Reviewer 3.

“Workers, especially, essential and/or healthcare workers, whose perceived health might have been badly affected by the pandemic, are mostly recommended for informing of the evidence about the association of social capital and locus of control with physical and mental health.”

Reviewer 4 Report

All countries had to deal with a pandemic that appeared quite recently, but its destructive impact is observed so far. the pandemic damaged the physical health of many people, but lockdown developed a wide variety of mental disorders. many of the psychological symptoms of a pandemic such as depression will continue to be seen in society in the years to come. The authors attempted to combine social capital and a sense of control with perceived health during the pandemic in Japan. the study was conducted on a representative group with the use of good psychological tools and appropriate statistical methods. The article is an example of how good methodology, advanced statistical tools combined with good theoretical preparation, allow you to create a short but concise and good-quality article.

I present some comments for authors to consider

In 1. Introduction - only one study on social capital and health was presented. Still other studies are worth citing. e.g:https://doi.org/10.3390/ijerph17196985

2. Materials & Methods

This part is described very well and does not require improvement. Noteworthy is the introduction by the authors: "2.2.4. Potential confounders"

Please indicate whether the test was carried out using an electronic form or using the paper-pencil method.

3. Results

4. Discussion

The average age of people participating in the study is higher than the average age of people in Japan. This issue should be raised in the discussion as it may be related to a medical conditio.

it is worth recalling the data showing that the social capital in Japan is very high, e.g.:

https://solability.com/the-global-sustainable-competitiveness-index/the-index/social-capital

Author Response

Response to the comments by Reviewer 4.

  1. Introduction: According to the comments by the Reviewers 2, 3, & 4, a total of ten references were additionally cited in Introduction, which were cited in Discussion of the previous manuscript. The following sentences were added in Introduction,

“Several studies have indicated that social capital influenced mental health during the COVID-19 pandemic.9-14 For instance, Budimir et al.14 indicated from a cross-sectional study in Austria during the pandemic that a higher level of social support was negatively associated with perceived stress, depression, anxiety, and insomnia.”

“Several studies have indicated that locus of control influenced mental health during the COVID-19 pandemic.17-20 For instance, Krampe et al.20 showed from a cross-sectional study in Norway and Germany during the pandemic that a higher level of internal locus of control was negatively associated with general mental distress, but an external locus of control was positively associated with general mental distress.”

The suggested reference (e.g.: https://doi.org/10.3390/ijerph17196985) is the same as Reference #18.  

  1. Materials & Methods: As previously stated in the section of Statistical methods, SAS statistical software was used for all analysis, and it means that the test was carried out using an electronic form.
  2. Discussion: Because we are not sure whether the average age of people participating in the study is higher than the average age of people in Japan, we do not discuss it. Thank you very much for introducing us the data showing that the social capital in Japan is very high (e.g.:https://solability.com/the-global-sustainable-competitiveness-index/the-index/social-capital). However, we chose not to add this URL in Discussion, because it is not directly relevant to our findings.

Round 2

Reviewer 1 Report

The authors did not address the issues raised by this review. Simply adding a paragraph on potential limitations is not appropriate and neither sufficient. Many essential factors that contribute to the perceived health must be included in the analysis to ensure the data interpretation is meaningful. I don't believe the quality of this manuscript in its current form meets the standard of our journal.

Author Response

As previously responded to the comments by Reviewer 1, we did not include the socio-environmental factors in the questionnaire, because participants might be sensitive to them, and we were concerned that such questions could reduce a response rate. We have added the sentences explaining this limitation in Discussion. However, the reviewer thought that adding a paragraph on potential limitations is not appropriate and neither sufficient. Accordingly, we found it impossible to address the comments in the review reports by Reviewer 1.